# Photothermal Effect of Gold Nanoparticles as a Nanomedicine for Diagnosis and Therapeutics

**DOI:** 10.3390/pharmaceutics15092349

**Published:** 2023-09-19

**Authors:** Panangattukara Prabhakaran Praveen Kumar, Dong-Kwon Lim

**Affiliations:** 1KU-KIST Graduate School of Converging Science and Technology, Korea University, 145 Anam-ro, Seongbuk-gu, Seoul 02841, Republic of Korea; p4praveen@korea.ac.kr; 2Department of Integrative Energy Engineering, Korea University, 145 Anam-ro, Seongbuk-gu, Seoul 02841, Republic of Korea; 3Brain Science Institute, Korea Institute of Science and Technology (KIST), 5, Hwarang-ro 14-gil, Seongbuk-gu, Seoul 02792, Republic of Korea

**Keywords:** nanomedicine, gold nanoparticles, photothermal effect, photonic PCR, cancer immunotherapy

## Abstract

Gold nanoparticles (AuNPs) have received great attention for various medical applications due to their unique physicochemical properties. AuNPs with tunable optical properties in the visible and near-infrared regions have been utilized in a variety of applications such as in vitro diagnostics, in vivo imaging, and therapeutics. Among the applications, this review will pay more attention to recent developments in diagnostic and therapeutic applications based on the photothermal (PT) effect of AuNPs. In particular, the PT effect of AuNPs has played an important role in medical applications utilizing light, such as photoacoustic imaging, photon polymerase chain reaction (PCR), and hyperthermia therapy. First, we discuss the fundamentals of the optical properties in detail to understand the background of the PT effect of AuNPs. For diagnostic applications, the ability of AuNPs to efficiently convert absorbed light energy into heat to generate enhanced acoustic waves can lead to significant enhancements in photoacoustic signal intensity. Integration of the PT effect of AuNPs with PCR may open new opportunities for technological innovation called photonic PCR, where light is used to enable fast and accurate temperature cycling for DNA amplification. Additionally, beyond the existing thermotherapy of AuNPs, the PT effect of AuNPs can be further applied to cancer immunotherapy. Controlled PT damage to cancer cells triggers an immune response, which is useful for obtaining better outcomes in combination with immune checkpoint inhibitors or vaccines. Therefore, this review examines applications to nanomedicine based on the PT effect among the unique optical properties of AuNPs, understands the basic principles, the advantages and disadvantages of each technology, and understands the importance of a multidisciplinary approach. Based on this, it is expected that it will help understand the current status and development direction of new nanoparticle-based disease diagnosis methods and treatment methods, and we hope that it will inspire the development of new innovative technologies.

## 1. Introduction

The recent advancements in nanotechnology and nanomaterials have led to several breakthrough technologies in nanomedicine for the early diagnosis and treatment of various diseases [1,2]. Metallic nanoparticles such as gold nanoparticles (AuNPs) [3,4,5,6], silver nanoparticles (AgNPs) [7], iron [8], zinc oxide [9], and titanium dioxide nanoparticles [10,11] have been extensively investigated for the applications based on the unique physical and chemical properties. Among metallic nanoparticles, AuNPs possess excellent synthetic versatility, biocompatibility, surface modification properties, and, most importantly, unique optical properties [12,13]. The optical properties of AuNPs are largely dependent on the size and shape, which are tunable from the visible to near-infrared (NIR) regions of light.

The diverse areas of applications with AuNPs in nanomedicine are shown in Figure 1 [14,15,16,17,18]. The size and particle-distance-dependent color changes in AuNPs have opened new avenues of applications for in vitro diagnostics, e.g., colorimetric assay with the naked eye to detect DNA targets or small molecules [14,19,20,21], lateral flow assay with the strong and stable color of AuNPs, and robust conjugation of antibodies for rapid kits that utilized for pregnancy test and COVID-19 test [22,23]. The strong scattering and optical properties of AuNPs have been utilized for in vivo imaging, e.g., photoacoustic imaging with a strong photothermal (PT) effect of AuNPs [17,24,25], strong scattering property of X-ray for computed tomography imaging agent, and surface-enhanced Raman scattering for optical imaging [16,26]. As a therapeutic carrier, various molecules such as small drugs, genes, or antibodies can be conjugated with AuNPs via strong Au-S conjugation chemistry to be delivered to their targeted sites [18,27,28,29,30]. Furthermore, the optical property of AuNPs is utilized for light-responsive therapeutic applications, e.g., photothermal therapy (PTT), which utilizes light to generate strong thermal energy for cancer hyperthermia [31,32,33], and photodynamic therapy (PDT), which utilizes AuNPs as photosensitizers or enhancer for an accelerated generation of reactive oxygen species [34].

Among various properties of AuNPs [35,36], the strong PT effects have been utilized for a diverse range of diagnostic and therapeutic applications such as photoacoustic (PA) imaging, photonic PCR, and hyperthermia, which opened a new era of applications in nanomedicine [37,38,39,40]. Therefore, in this review, we focus on the basic principles of the PT effect and the recent advances in diagnostic and therapeutic applications of AuNPs.

## 2. Basics of the Photothermal Effect

### 2.1. Mechanism and Nanostructures

When the size of plasmonic metal nanoparticles is small, the absorption process of light is more dominant than the scattering process [41]. Absorbed light energy is converted into thermal energy through a thermal relaxation path and then released to the outside. The hot carriers generated by light interaction can participate in an external chemical reaction or disappear through rapid internal relaxation. The relaxation process is performed in the order of Landau damping, carrier relaxation, and thermal dissipation as illustrated in Figure 2. The absorbed light energy generates a strong electromagnetic field around the nanoparticles through plasmon resonance on the surface of the metal NPs, which makes free electrons vibrate rapidly, causing excitation of energetic hot electron–hole pairs, and the excited surface plasmon undergoes fast decaying nonradiative Landau damping (1–100 fs) [42,43]. The initially excited electrons are rapidly equilibrated through electron–electron scattering, and therefore, the distribution of electrons forms a “hot” Fermi–Dirac distribution. Then, through the coupling between the hot electrons and the phonons in the metal lattice, the hot thermal distribution is cooled and the lattice temperature rises [41].

The PT performance of AuNPs can be tuned from the visible to the NIR region of light by varying their size and shape (Figure 3a–f) [44,45]. The light absorption by body tissue could be minimized in the wavelength at NIR-I and NIR-II (Figure 3a). Spherical AuNPs possess a surface plasmon resonance (SPR) at 520 nm (Figure 3b), which becomes broader and slightly red-shifted when the particle size increases up to 100 nm, indicating the limited SPR property of spherical AuNPs to NIR wavelength. By contrast, gold nanorods (AuNRs) can easily attain the SPR property at 800 nm and show excellent tunability by changing the aspect ratios (Figure 3c,d). Gold nanoshells (AuNSs) and gold nanocages (AuNCs) also possess NIR absorption properties (Figure 3e,f). It should be noted that extinction is the sum of contributions for absorption and scattering, which is significantly governed by the dimensions of materials. A higher contribution of absorption is important for realizing the higher PT effect.

### 2.2. Enhancing Photothermal Performance with New Material Design

The photothermal conversion efficiency (η) of the plasmonic materials can be calculated using the following Equation (1) [46]: (1)$$\eta =\frac{\mathrm{hS}\;(T_{\max}-T_{\mathrm{amb}})-Q_{\mathrm{dis}}}{I(1-10^{-A})},$$
Q_dis_ = 103 m C ∆T/t,(2)
where *h* and *S* denote the heat transfer coefficient and surface area of the container in which NPs are irradiated, respectively, T_max_ and T_amb_ are the maximum equilibrium temperature and ambient temperature of the surroundings, respectively, I is the laser power, and A is the absorbance of the NPs at the given wavelength. Q_dis_ (mW) represents the laser-induced heat in the container, which can be calculated using Equation (2). Recent reports are summarized in Table 1 showing how the photothermal conversion efficiency of various AuNPs can be tuned by varying their dimensions and the intensity of laser sources [5,47]. The PT conversion efficiencies of AuNPs can be varied from 22% to 103% by varying their shape and size. Several studies have suggested that AuNRs are a better candidate than AuNPs for PT applications. Gold nanoflower (AuNF)-type structures possess 74% PT conversion efficiency owing to their strong SPR absorption in the NIR region, whereas gold nano bipyramids (AuNBPs) have shown 95% PT conversion efficiency, indicating the structural characteristics of AuNPs for photothermal studies.

In addition to the change in dimensions, the addition of nanoscale materials that can improve light absorption or the heat transfer process is an alternative way to improve photothermal conversion efficiency. As shown in Figure 4, graphene coating on AuNPs [59,60,61,62], silica shell [63], and carbon nanotube with Au shell [64,65] are the ways to obtain higher PT effect and improved PA signal intensity [66,67]. Gao et al. showed that graphene oxide coating improved the stability and PT conversion efficiency of AuNPs [59]. Studies have shown that irradiation of this hybrid system using 808-nm laser (0.3 W cm^−2^, 10 min) resulted in a significant rise in the temperature (ΔT = 23.00 °C) compared to the control nanomaterials without a graphene oxide layer. He et al. reported that graphene oxide-coated Au nanosheets exhibited a PT conversion efficiency of 30%, which was higher than that of AuNR (21%) [60].

Because reduced graphene oxide (r-GO) possesses delocalized pi electrons in contrast to nonreduced graphene oxide, it showed an improved light absorption property. Lim et al. showed that coating AuNRs and AuNSs with r-GO resulted in a 2.9-fold enhancement in PT efficiency using NIR light compared to uncoated or nonreduced graphene oxide-coated materials (Figure 4a–d) [61]. The enhanced PT performance could also improve PA signal intensity in in vivo and in vitro conditions (Figure 4e). Studies showed that a chemically reduced GO layer using hydrazine exhibited a PA signal amplitude of 53.77 ± 1.71 at 6.2 mJ/cm^2^, which is two times higher than the signal intensity for AuNRs (Figure 4f,g) [62]. The presence of silica shells can also improve the heat transfer process between AuNPs and the surrounding medium by reducing the interfacial resistance between them. Yang et al. demonstrated a relationship between silica shell thickness and their PT conversion efficiency in AuNRs. They reported an optimum shell thickness of 20 nm to achieve the highest PT performance (Figure 4h–j) [63].

Carbon nanomaterials are well known for their enhanced light absorption and PT conversion efficiencies. Song et al. showed that the incorporation of carbon nanotube increased the extinction coefficient of AuNPs up to 120-fold with an increment in temperature up to 70 °C in 5 min under irradiation by an 808 nm laser source [64]. The increment in the PT effect of AuNPs was due to the enhanced interparticle coupling of the SPR peaks inside the carbon nanowell and resulted in maximum absorption of light for the PT conversion with improved photostability.

Kim et al. showed that gold-plated carbon nanotubes (GNTs) with an inner diameter of 1.5–2 nm and gold thickness of 4–8 nm showed 10^2^-fold enhancement in signal intensity as a PT material for photoacoustic imaging applications (Figure 4k,l) [65]. To measure the PT conversion efficiency, the PA signal intensity and bubble-formation thresholds of GNTs were compared with those of AuNPs, AuNR, AuNSs, and carbon nanotube (SWNTs); results indicated that GNTs showed a higher PA signal intensity and lower bubble-formation threshold compared to AuNR, AuNPs, and SWNTs (Figure 4m,n).

Utilization of polymeric materials such as polydopamine and polypyrrole is also a possible design strategy for enhancing the PT effect in AuNPs. Li et al. showed that polydopamine-coated Au nanostars (AuNSTs) with an average size of 90 nm exhibited an increase in temperature up to 55 °C using a laser source of 808 nm (2 Wcm^−2^) for 10 min and improved tumor ablation in Hela cells up to 90% [66]. Cao et al. reported that polypyrrole-coated AuNR showed a final temperature of 58.3 °C compared to the uncoated AuNR along with tumor suppression, and the antitumor property could be further improved by loading doxorubicin (DOX) into the polymer shell. The incorporation of Fe_3_O_4_ into the core enhanced the therapeutic application as demonstrated by improved magnetic resonance imaging and CT imaging efficiencies [67]. Incorporating other semiconducting metal ions into the AuNPs induces a redshift in their SPR peak to the NIR region [68,69]. The strong SPR absorption in NPs can enhance the charge separation in semiconducting materials, which results in the conversion of optical energy into thermal energy. Leng et al. showed that incorporation of copper sulfide into AuNRs induced an SPR peak shift up to 803 nm compared to bare AuNR and increased the light-to-heat conversion efficiency from 39% to 56% [68]. Moreover, the copper coating enhanced the photostability of the hybrid system to afford a final temperature of 60 °C under an 808 nm laser (0.9 W·cm^−2^) for 10 min compared to AuNRs, which showed a decrease in temperature from 52.5 °C to 47 °C after several cycles. Following this work, Kumar et al. showed that molybdenum coating improved the PT effect and a final temperature of 60.3 °C could be attained for the AuNBP-Mo hybrid material [69]. Compared to molybdenum disulfide and AuNBP, the hybrid material showed a decrease in Hela cell viability up to 21.6%, indicating the enhanced PTT effect by the AuNBP-Mo hybrid system.

These results indicate that materials design at the nanoscale is also a viable way to obtain improved PT performance by improving the light absorption property and heat transfer process. In addition, the colloidal stability and incorporation of chemical functionality could also be improved.

## 3. Diagnostic Applications

### 3.1. Photoacoustic Imaging

Photoacoustic imaging (PA) is a medical imaging technique utilizing the photoacoustic properties of light-absorbing molecules. PA imaging offers a high spatial resolution of less than 5 μm and a deeper imaging depth of up to 6 cm compared to other optical imaging techniques. PA imaging is based on the difference in optical properties of biological tissues, whereas ultrasound-based imaging depends on their mechanical properties. Thus, PA imaging produces a higher contrast for the imaging application than ultrasound-based methods [70]. Various contrasting agents can be used for PA imaging applications, which can either be endogenous, such as melanin and hemoglobin, or exogenous, such as fluorophores, dyes, and nanomaterials. Because of the low concentration of biomolecules, an exogenous contrasting agent is utilized for better imaging modality [71]. Contrasting agents generate acoustic signals through the thermoelastic expansion of tissues by the absorption of light and produce broadband sound waves. These acoustic waves are detected by an ultrasound wave transducer that converts the sound waves into PA signals and then into images depending on their signal arrival time (Figure 5) [72,73].

The amplitude of PA signal intensity can be calculated using the following Equation (3) for the initial pressure (P_0_) rise due to the thermal expansion of tissues [74]:P_0_ = Γη_th_μ_a_F,(3)
where Γ is a dimensionless quantity known as the Grüneisen parameter, which is a measure of the PA signal amplitude, η_th_ represents the heat conversion efficiency of the photon-absorbing species, μ_a_ denotes the optical absorption coefficient of the contrasting agent, and F is the local optical fluence. Thus, from Equation (3), it is clear that the PA signal amplitude can be controlled by the optical absorption coefficient and the heat conversion efficiency of the contrasting agent.

Current challenges in fabricating PA contrasting agents are dependent on the absorption and scattering properties of the contrasting materials, their tissue penetration capabilities, background noise from the tissues, and the attenuation of the light source. Among the various contrasting agents available, the strong NIR (I and II) absorption coefficients and the minimal scattering properties of AuNPs make them a better candidate for PA imaging applications [71]. A variety of AuNP-based nanostructures such as AuNRs [75,76], AuNSTs [77], AuNCs [78], and AuNSs [79] have been commonly employed for PA imaging. According to the Rayleigh scattering principle, the absorption in the higher-wavelength region (NIR-II) degrades the scattering properties of the AuNPs and can enhance the PA signal intensity. Compared to other contrasting agents, AuNPs possess excellent photostability and photobleaching properties.

Because of their low absorption wavelength region of approximately 520 nm, spherical AuNPs are not used directly as a PA imaging agent, and blood components also mostly absorb in the same wavelength region. Therefore, distinguishable spectral and anatomical features are difficult with spherical AuNPs. However, the hierarchical assembly of AuNPs can tune the SPR band absorption to the NIR region by the strong plasmonic coupling between the adjacent particles [80,81]. Han et al. fabricated 5-nm ultrasmall AuNPs, and their clustering property inside the cancer cells improved their absorption coefficient for PA imaging applications [80]. Functionalization of these ultrasmall AuNPs with the epidermal growth factor receptor-targeting ligand enhanced their target specificity and demonstrated that the 5-nm AuNPs could act as a better PA imaging agent than 40-nm AuNPs, with high penetration and renal clearance properties (Figure 6a). Cheng et al. showed that spherical AuNPs could be aggregated using the enzyme furin under controlled pH conditions, and these aggregates could be used as a contrasting agent for the PA imaging of HCT-116 tumor-bearing mice [81]. At pH 5.5, the absorption peak shifted from 520 to 700–900 nm with an increase in temperature up to 63.4 °C compared to the AuNPs without furin. Meanwhile, structural engineering of spherical AuNPs is an adopted strategy for the fabrication of PA imaging agents with high absorption cross sections. Lu et al. prepared 45-nm hollow spherical AuNPs with an absorption window in the NIR region (800 nm). The hollow AuNPs showed enhanced photoacoustic properties in nude mice with improved clarity for brain vasculature after 2 h of intravenous administration [82]; they exhibited absorption and scattering coefficients of ∼540 and ∼78 μm^−1^, respectively, and the excellent absorption coefficient made these nanoparticles a better PA imaging agent for brain blood vessels in mice as small as 100 μm in diameter.

Nguyen et al. prepared a chain-like assembly of AuNPs using two different peptide templates, Cys-Ala-Leu-Asn-Asn, and cysteamine, for PA imaging applications [25]. The chain-like assembly possesses an absorption band at 650 nm with an average diameter of 20 nm. Conjugation of these nanoparticles with polyethylene glycol (PEG) and arginylglycylaspartic acid (RGD) peptide improved blood circulation and target specificity. The PA contrasting ability of this nanoparticle conjugate was studied in Hela cells, where the cells treated with AuNPs showed enhanced PA signal intensity compared to the untreated cells, with a PA signal of 152.56 ± 18.71 (a. u.) for cells vs. 5.47 ± 0.50 (a. u.) for the background. The present nanosystem has been studied for PA imaging of retinal tissues owing to its salient structural features compatible with those of retinal tissues.

AuNRs are widely employed as a PA imaging agent because of their excellent SPR and PT properties that depend on their different aspect ratios in length and width and their ease of synthesis. Chen et al. showed that a miniature-sized AuNR [(8 ± 2) nm × (49 ± 8) nm] can absorb light in the NIR-II region and showed 3.5 times enhanced contrasting ability for PA imaging compared to larger AuNRs (Figure 6b) [83]. These miniature-sized AuNRs absorb at 1064 nm with a size almost 5–11 times less than that of commonly used AuNRs for PT and PA applications. In vivo studies showed that inside the tumor tissues, the AuNRs showed 4.5 times enhancement in the PA signal; thus, this work highlighted the importance of the structural aspects of AuNRs with respect to their optical absorption window. The PA signal intensity of the small and large AuNRs were compared after their injection into tumor-bearing mice, and studies showed that both these AuNRs exhibited target specificity after conjugating with GRPR-targeting peptides and Cy5 dyes. Laser irradiation studies showed that after 24 h of injection, the nontargeted large AuNRs displayed a higher PA signal intensity than the small AuNRs owing to the tumor heterogeneity (Figure 6b(ii,iii)), whereas for the target-specific AuNRs, the smaller AuNRs showed an enhanced PA signal intensity than the larger AuNRs [Figure 6b(iv,v)]; this result highlights the importance of the size aspect for target specificity and PA signal intensity. Furthermore, Knights et al. demonstrated that among the various AuNRs synthesized with similar aspect ratios, the 40- and 50-nm AuNRs showed the highest toxicity in lung cancer cells with a concentration of 3 × 10^10^ NP mL^−1^, whereas AuNRs with 10-nm size showed the highest PA signal intensity with less toxicity, indicating the importance of size and concentration of AuNRs for PA studies [84].

Silica coating of AuNRs improves PA signal intensity by providing stability and plasmonic coupling to AuNRs. Chen et al. showed that 20-nm silica shell coating improved the PT and PA signal intensity of PEG-AuNR (34 × 9 nm) compared to CTAB AuNR and PEG-AuNR without silica shell [85]. Xu et al. showed that mesoporous silica shell coating not only improved the PA signal intensity but also enhanced the therapeutic application of AuNRs by enhancing DOX loading efficiency for chemotherapy in breast cancer cells [86]. Furthermore, a bacteria-type therapeutic agent (SiO_2_@AuNR-PEG) was fabricated with a surface area of 470 m^2^/g and pore size of 4–8 nm, which enabled the enhanced conjugation of DOX (40.9%, *w*/*w*) (Figure 6c). Chelating this therapeutic agent with ^89^Zr using PEG chains afforded simultaneous positron emission tomography (PET), PA, and PT chemotherapy applications in 4T1 murine breast cancer-bearing mice. Under a pH of ~5.5 and laser irradiation (808 nm, 0.5 W/cm^2^) 55.5% of DOX was released with 4.5-fold enhancement in PTT compared to the control nanomaterials. As shown in Figure 6c(vi), the SiO_2_@PEG-AuNR therapeutic agent exhibited a 4.7-fold enhanced PA signal intensity after 24 h compared to the pre-injection in tumor-bearing mice as shown in yellow circles and enhanced PET imaging after 24 h (Figure 6c(vii)). The yellow colored arrow heads in Figure 6c(vii) indicates the tumor in mice. Because of the strong NIR absorption, photostability, and PT conversion efficiency of the theragnostic system, it can be used for multimode imaging and PTT studies of tumors.

**Figure 6 pharmaceutics-15-02349-f006:**
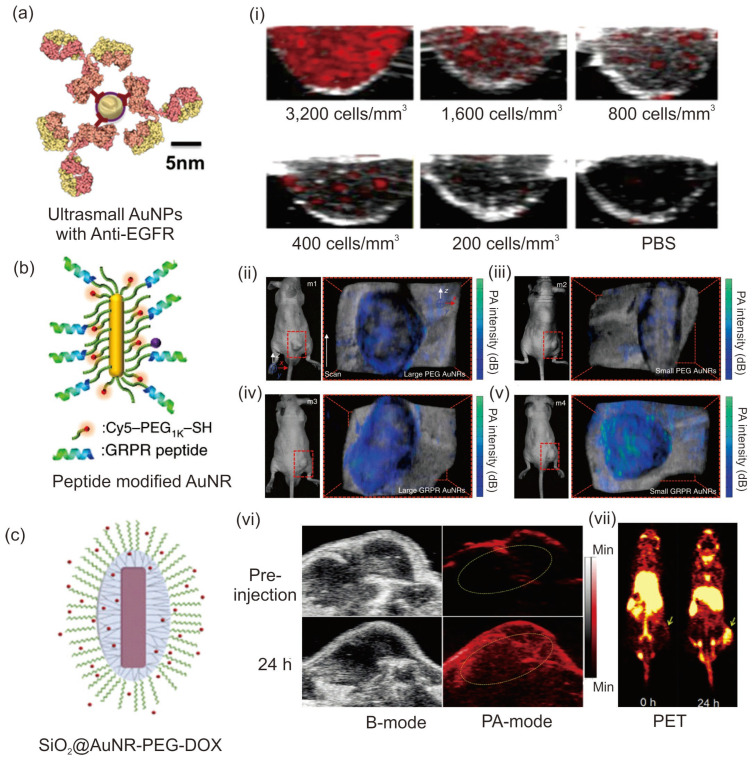
Application of various AuNPs in PA imaging. (**a**) Schematic of 5 nm ultrasmall AuNPs for PA imaging application and (**i**) cross-sectional images of A431 cells labeled with 5 nm maps for 10 h at different concentrations. (**b**) Schematic and photographs of photoacoustic imaging of tumor-bearing mice from four different mice samples (m1–m4) using large and miniature AuNRs. (**ii**,**iii**) PA imaging for the nontargeted larger and small AuNRs, respectively; (**iv**,**v**) PA imaging of the GRPR-targeted large and small AuNRs, respectively. The colored maps, which represent the PA imaging signal intensity, are overlayed with the ultrasound images to provide anatomical information. (**c**) Schematic of the prepared SiO_2_@PEG-AuNR with DOX. (**vi**) In vivo PA imaging of 4T1 tumor-bearing mice before and after 24 h post-injection. (**vii**) PET imaging of 4T1 tumor-bearing mice at different time points. Reproduced with permission from [80,83,86]. Copyright 2019, Optical Society of America, Springer Nature 2019, and Elsevier 2018.

Thus, it can be concluded from the existing reports that AuNPs can be used as a better contrasting agent for PA imaging studies. However, owing to the difference in the biological environment and deep tissue penetration properties, optimizing the absorption and scattering properties of AuNPs is still challenging. This can be overcome rapidly by the advancements in nanotechnology and more versatile synthetic procedures for preparing stable and optically controlled AuNPs for PA imaging applications.

### 3.2. Plasmonic PCR-Based Methods for Nucleic Acid Detection

PCR, which was invented by Kary Mullis in 1983 [37,38], is a technique used for the detection of nucleic acid targets. In this technique, three distinct thermal cycles are required at three different stages, namely denaturation, annealing, and extension, to amplify the target DNA. The plasmonic PCR uses a light-induced thermal method instead of a conventional Peltier system. Many NP-based materials (e.g., AuNPs, Au nanofilm, and Fe_3_O_4_) are used for thermal amplification. Owing to the PT properties of Au, it was employed as a thermoamplifier in the photonic PCR methods compared to other nanomaterials. The PT properties of AuNPs can improve the thermocycling property in three ways. First, the incident light can be utilized to produce the required local heat energy. Second, it enhances the heat transfer between the solution and analyte, Third, the increased volumetric heating is possible due to the large surface contacts. Therefore, the use of metallic photonic PCR agents can improve the heating and cooling cycles more than conventional PCR techniques.

For PCR, the strong absorption property of water for NIR light >1000 nm and heat production by plasmonic nanomaterials are utilized. AuNPs can be simply dispersed into the water for the plasmonic PCR as a thermal amplifier, which can provide a uniform heating rate compared to the Peltier system. Various Au nanostructures, such as AuNRs and AuNBPs, are commonly employed in photonic PCR.

In an earlier work, Roche et al. compared the thermocycling power of AuNPs with that of the conventional Peltier-based PCR systems for the amplification of human androgen receptor DNA [87]. The studies showed that isolated AuNPs (method) did not improve the thermocycling property, whereas a solution of AuNPs directly added into the PCR system (contact method) showed improved thermocycling property of roughly 30 PCR cycles in a short time of 10 min. To prevent the interaction between the polymerase and AuNPs, which can render the activity, bovine serum albumin (BSA) was added to the PCR system to stabilize the AuNPs. The optimization conditions for both AuNPs (6.6 pM) and BSA (10 ng/mL) for 25 μL of the total PCR system showed a heating rate of 7.62 °C s^−1^ and cooling rate of 3.33 °C s^−1^ using a 532 nm laser with a power of 2.7 W. Following this work, they suggested that using AuNRs instead of AuNPs, the cycling time could be reduced from 10 min to 54 s using an 808-nm laser (2 W) with a large reduction in PCR analysis time by 3 s compared to the conventional methods [88]. Li et al. showed that the thermocycling system could be combined with a microfluidic chip system, which eventually enabled a reduction in the total PCR volume in multiple channels, and consequently, the PCR amplification and thermal cycling rate increased with heating and cooling rates of 300 and ~75 °C s^−1^, respectively, using a 781-nm laser (13.6 mW) [89]. This microfluidic chip-based photonic PCR system allowed multiplex detection of DNA targets in a short time.

Because of the excellent uniformity in dimension and PT conversion efficiency of AuNBPs, they have been employed in plasmonic PCR applications [90,91]. Lee et al. utilized AuNBPs coated with PEG as a thermal amplifying material (Figure 7a,b), and further modification of these NPs using silica shell coating improved stability and suppressed the interaction of AuNBPs with the polymerase agent [90]. Moreover, silica shell coating enables heat dissipation and acts as a spacer between the fluorescent dye SYBR Green I dye and AuNBPs for amplification. Figure 7c,d presents the observed thermal cycling amplification of the system with varying concentrations of AuNBPs by varying the optical density (OD) of the sample using a blue light-emitting diode (LED). Studies showed that an optimum OD of 21.5 enabled a reduction in thermocycling time, with heating and cooling rates of 16.6 ± 2.4 and 9.4 ± 0.8 °C s^−1^, respectively, and a further increase in OD does not induce any change in the amplification rate (Figure 7c). As a practical application, the amplification of the Viral M13 amplicon was studied, and using the SYBR Green I dye, the emission at 520 nm was studied with the amplification cycle (Figure 7e,f). Results showed that the plasmonic PCR method could finish 40 cycles within 7.5 min. The designed plasmonic platform further extended its application for the rolling circle amplification of target DNA and multienzyme amplifications.

For quantitative PCR (qPCR) techniques, Au thin films and doping methods are other strategies that have been adopted recently. In the doping method, AuNPs can be dopped with the amplification PCR solution to enhance the amplification and facilitate a reduction in the amount of AuNPs; however, one drawback of this technique is the unwanted interaction between the NPs and primers [92,93]. In an Au nanofilm-based platform, within a short period, a hot plate can be generated, which reduces the amplification time. Son et al. fabricated an Au nanofilm on polymethyl methacrylate microwells using different Au-deposition thicknesses from 10 to 120 nm and coated the Au films with polydimethylsiloxane to avoid the unwanted interaction and deposition of primers and polymerase on Au [94]. With a blue LED (450 nm, 890 mW), the maximum heating and cooling temperature were achieved with 30 thermal cycle amplification within 5 min. The temperature can be increased from 55 °C (annealing) to 95 °C (denaturation) with heating and cooling rates of 12.79 and 6.6 °C s^−1^, respectively, indicating that this Au nanofilm could be compared with the traditional PCR method for three different temperatures for denaturation, annealing, and extension (i.e., 94 °C, 60 °C, and 72 °C, respectively). To improve the thermal amplification efficiency of the system, two parallel Au nanofilms were placed in the polymer microwells in later studies [95]. The amplification of cDNA and λ DNA could be performed with high accuracy and reproducibility with this Au nanofilm-based PCR method.

Jalili et al. showed that a plasmonic PCR system using an Au nanofilm (120 ± 5 nm) coupled with a PDMS microfluidic chip reduced the amplification time for alcohol oxidase gene detection [96]. The Au nanofilm could attain 30 thermocycles from 60 °C to 95 °C within a short time of 13 min. To avoid the evaporation of sample drops in the microfluidic chips, mineral oils were used, and using tapes, the adhesion between the nanofilm and PDMS fluidic system could be enhanced. To improve the thermal cycling property, two consecutive LEDs were used to irradiate the Au film, and average heating and cooling rates of 7.37 ± 0.27 and 1.91 ± 0.03 °C/s, respectively, could be achieved. The Au film could amplify the PCR detection by 30 thermal cycling repetitions with a deviation of ±0.41 °C from set points, suggesting the accuracy of the method. Kang et al. used Au nanoislands as a photonic thermal amplifier, where conjugation with microfluidic channels induced a rapid and sensitive plasmonic PCR amplification for lambda DNA within 264 s over 40 thermal amplification cycles and showed a PCR response for the SARS-CoV-2 envelope protein within 306 s [97]. Au nanoislands can control the heating and cooling rates to 11.95 and 7.31 °C/s, respectively, which is faster than the available benchtop PCR techniques with an amplification efficiency of 91%. Thus, the photonic PCR techniques in conjugation with the microfluidic system can enhance the PT amplification cycle, reduce assay time, and can be used for point-of-care applications for viral and bacterial detection.

Recent studies using plasmonic PCR techniques have revealed that AuNPs can be used as an amplifier to improve the thermal amplification cycle in PCR methods in a shorter time. The conventional PCR system uses thermocouples and the Peltier system for the amplification of target DNA. However, using plasmonic materials such as AuNPs, light energy can be completely converted into thermal energy, and the amplification rate is higher than that of conventional Peltier systems. Moreover, the multiplexing capability and the cost of the current assay is a limiting factor, whereas plasmonic-based PCR techniques can overcome most of these issues in conjugation with other platforms such as microfluidic channels and thin-film techniques. Thus, existing studies elucidate that AuNRs and AuNBPs have shown excellent thermal amplification properties compared to spherical AuNPs. Au nanofilms or nanoislands can create a homogenous and uniform thermal amplification within a short period.

Although the plasmonic PCR technique overcomes many of the issues, one of the biggest problems with plasmonic PCR systems is the unwanted interaction between the targets and polymerase agents. Many of the reagents and functionalities in the target DNA can induce a strong bond with the plasmonic materials, which affects the performance and leads to incorrect analysis. Therefore, special care must be taken to control these adverse effects.

As a future perspective and direction toward plasmonic PCR-based methods, one can envisage preparing various hybrid materials from inorganic and organic PT materials to reduce the number of metal NPs. The PT conversion property of polydopamine, BODIPY molecules, phthalocyanine, etc. can be incorporated with AuNPs for future directions. Another strategy is to reduce the volume of the PCR solution, which can be rectified using various techniques in combination with the PCR method, such as microfluidic channels and nanofilms. Moreover, controlling the heating and cooling rates is also an important factor in the fabrication of plasmonic PCR-based platforms. In the near future, myriad plasmonic PCR-based systems are expected to resolve existing issues associated with conventional PCR techniques for the diagnosis of various diseases and NA amplification.

## 4. Therapeutic Application of AuNPs

Because of their excellent biocompatibility, AuNPs have been used extensively for the treatment of various diseases [98,99]. AuNPs have been investigated as a carrier of drug molecules of interest. AuNPs could be used as a therapeutic agent for cancer and arthritis as an anti-inflammatory agent owing to their enhanced-permeation-and-retention effect [13]. For the target-specific delivery of drugs and treatment of tumors, the surface of AuNPs can be functionalized with several antigens and antibody receptors, and they have shown excellent therapeutic applications. Thiol chemistry on Au is the commonly employed method for surface modification of AuNPs with various target-specific agents such as folic acid and anticancer drugs. The AuNPs improved the solubility and internalization of DOX, which is otherwise difficult owing to the low solubility of the drug. The target-specific delivery of the drug resulted in an improved anticancer therapeutic effect [100].

Furthermore, the PTT and PDT of AuNPs are well-known methods in anticancer therapy [101]. AuNPs can improve the ROS formation in tumor cells for cancer cell ablation or act as a carrier of photosensitizers toward cancer target cells for photodynamic therapy studies. It is possible to use a NIR wavelength to deliver light energy to destroy the cancer cells in the deep tissue. For example, spherical AuNPs, which exhibit an absorption band at 520 nm, cannot be used for deep tissue therapy, whereas anisotropic structures such as AuNRs, AuNCs, and AuNSTs can be used for deep tissue PTT [102]. In this regard, AuNPs have been widely investigated as a nanomedicine for anticancer therapy. Beyond chemotherapy, immunotherapy is now widely adopted in the clinic as a promising therapeutic. For immunotherapy, AuNPs can play an important role as a carrier of biomolecules to improve the efficacy of immunotherapy. In addition, the PT effect of AuNPs can synergistically improve cancer immunotherapy by controlling the light dose.

### 4.1. Conventional Hyperthermia Using AuNPs

PTT is a noninvasive method using the PT conversion ability of AuNPs by absorbing light for tumor ablation studies. Excitation of AuNPs using a suitable light source results in the dissipation of heat to the surrounding medium, and eventually the temperature in the tissues gradually increases to 45 °C–55 °C, followed by an increased blood flow and homeostatic condition with irreversible cell damage either by apoptosis or necrosis [103]. For an effective PTT process, AuNPs should first localize at the tumor site, which can be achieved using suitable surface-modifying agents for better circulation and site-specificity of AuNPs, and then the suitable choice of light source for tumor ablation without harming healthy tissues. Many research works have accordingly been undertaken in recent years on the structure and shape of AuNPs for effective PTT. AuNPs, which can absorb light in the NIR-I and NIR-II regions (750–1000 and 1000–1700 nm, respectively) are interesting because the water absorption is less in this window and deep tissue penetration can be achieved using such AuNPs. Various nanostructures such as AuNRs [104,105], AuNCs [106,107], and AuNSTs [108,109] have recently been reported for PTT applications in the NIR region.

Sun et al. showed that the temperature-induced assembly of AuNPs in conjugation with an elastin-type polypeptide (Val-Pro-Gly-Xaa-Gly) (ELP) resulted in a necklace-type assembly, and these assembled ELP-AuNPs showed an enhanced NIR absorption and PT effect compared to spherical PEG-AuNPs [110]. NIR laser irradiation using 808-nm laser showed a rise in temperature up to 60 °C with a PT conversion efficiency of ca. 30%, which was more than that of the PEG-AuNPs (21%). In vivo studies showed that administration and laser irradiation (808 nm, 1.5 W, 7 min) in tumor-bearing mice induced tumor ablation compared to the control, and the chances of tumor recurrence were completely suppressed. In another study, Wang et al. showed that silk fibroin-induced assembly of AuNPs can fine-tune the absorption of spherical AuNPs to the NIR window up to 808 nm and can act as a PTT agent for breast cancer cells with a short irradiation period of 6 min (4 W cm^–2^ for 6 min) [111]. The PT conversion efficiency of this assembly was found to be 38.42% more than bare AuNPs and reached a maximum temperature of 55 °C for the tumor ablation studies.

Conjugating AuNRs with various peptides, PEG, and target-specific receptors is a strategy adopted recently; such surface-modified AuNRs can enter tumor tissues by endocytosis or pinocytosis with minimal cytotoxicity [112,113]. Patino et al. fabricated an AuNR system that could specifically target the adenocarcinoma MUC-1 by conjugating it with nonimmunogenic antitumor-antibody-derived-1 peptide [105]. To improve the cell internalization, the AuNR was further conjugated with myristoylated polyarginine peptide, and studies revealed that the peptide-based AuNR showed an enhanced PTT effect compared to the control systems. Fluorescence microscopic studies showed the destruction of tumor cells, whereas there was no damage to the surrounding cells, indicating the target-specific internalization of the AuNR. Kang et al. prepared porphyrin-appended AuNR for PTT and imaging applications [114]. Porphyrin improves the therapeutics application and enables tracking of the AuNR owing to its fluorescence. For target-specific internalization, the AuNR is further conjugated with trastuzumab (TAuNRs) for targeting HER-2 overexpressing breast cancers. Using an NIR laser (808 nm, power density of 2 W/cm^2^) for 10 min, the temperature of TAuNR reached up to 63.8 °C, which is sufficient for tumor ablation studies. The PT ablation studies on BT474 and SK-BR-3 using TAuNR showed significant cell cytotoxicity than AuNR alone. The X-ray and fluorescence microscopic studies showed that TAuNR was more internalized in the BT474 breast cancer xenograft mouse than in the MDA-MB-231 xenograft mice, indicating the target specificity of TAuNRs.

The synergic effect of PTT and chemotherapy is a good strategy for cancer ablation studies. AuNRs conjugated with chemotherapeutic drugs cause less toxicity and reduce the dose of drugs needed for tumor ablation. Duan et al. prepared a chitosan-coated AuNR as a carrier for DOX and in vitro studies showed that the nanoparticle system can deliver DOX effectively and induce synergic chemo-PTT for tumors [115]. In another study, Wang et al. showed that polydopamine-coated AuNR loaded with thiolated tumor-homing peptides could act as a carrier for DOX, and pH-induced controlled release of the drug resulted in simultaneous chemotherapy and PTT in tumor cells [116]. Xu et al. showed that AuNR decorated with HA and folate receptors can induce the target-specific release of DOX into the tumor cells for synergic PTT and chemotherapy applications [117]. The cell viability of MCF-7 cells decreased by up to 31% through the internalization of AuNR and laser irradiation. The temperature of the AuNR system reached up to 67.5 °C with a 1.5-W/cm^2^ laser source, which resulted in the suppression and recurrence of the tumor.

Turcheniuk et al. reported that r-GO-coated AuNR showed excellent PT conversion efficiency for suppressing the growth of human glioblastoma astrocytoma (U87MG) cells in mice [118]. The AuNR was further conjugated with tat protein for better target specificity toward U87MG. Irradiation of the AuNR system using an 808-nm laser increased the temperature of the medium by 60 °C within 10 min and induced PT ablation in cancer cells. To find the target specificity in PTT studies, r-GO@AuNR-tat was further conjugated with fluorescein isothiocyanate dye, and the results were compared with those of non-tat-conjugated AuNR. Studies showed that r-GO@AuNR-tat was distributed in the entire cytoplasm compared to the non-tat r-GO@AuNR, indicating the physiochemical aspect of the nanoparticle system for internalization in U87MG. Furthermore, Lee et al. demonstrated that silica-coated AuNR-bearing nicotine receptor peptides showed improved blood circulation and photostability and inhibited the growth of glioma tumors. Studies showed that the PT conversion efficiency of AuNR was enhanced because of silica coating and reached up to 50 °C after only 5 min of NIR exposure [119].

A novel cancer theragnostic agent was fabricated by Liu et al., in which mesoporous silica coating on a tLyp-1 peptide-functionalized AuNR-bearing indocyanine showed improved PTT and photostability for imaging applications [120]. Duan et al. reported a 77% increase in PT efficiency and a final temperature of 63 °C in AuNR with silica shell coating compared to control AuNRs without silica coating [121]. To improve the biocompatibility and reduce the toxicity of CTAB in AuNR, AuNCs were also employed in the synthesis step. The AuNRs@SiO_2_@AuNCs were treated with human gastric cancer cell line MGC803, and a low-power laser source was used (808 nm, 2.0 W/cm^2^, 3 min); the survival rate for cancer cells was found to be only 39.3%, indicating the enhanced PT effect of this hybrid material. Because of the excellent fluorescence property of AuNCs, the hybrid material is used for cancer cell imaging studies, highlighting the importance of this multifunctional nanomaterial for image-guided PTT.

AuNSTs are excellent candidates for simultaneous PT and photoacoustic imaging applications [122,123]. The star shape with several tips on its surface possesses excellent PT conversion efficiencies, typically in the NIR window. A recent report showed that AuNSTs possess excellent PT conversion efficiencies (90–94%) than AuNSs and AuNRs. In in vivo studies, successive tumor ablation was observed in sarcoma tumor-bearing mice [109]. D’Hollander et al. reported that AuNSTs coated with PEG-maleimide showed high PTT efficiency along with high-contrast signal intensities for PA and CT imaging studies [124]. Because of the laser-induced stability issue and their structural deformations, silica shell coating is an adopted strategy for better PTT studies [125,126].

Although this is the scenario, toxicity, and biodistribution of AuNPs are still challenging and questionable for clinical trial experiments. Moreover, the prolonged use of AuNPs can damage DNA and cause the generation of various free radicals, which can change the signal-transduction efficiency of various cells. Moreover, repeated laser irradiation can damage the NP structure, which makes renal clearance difficult and can induce cell toxicity. The use of various biocompatible polymer coatings or the fabrication of hybrid nanomaterials is a solution to these problems.

### 4.2. AuNPs for Cancer Immunotherapy

#### 4.2.1. AuNPs as a Nanocarrier for Antigens and Adjuvants to APCs

Antigen-presenting cells (APCs) act as a bridge between innate or the reprogramed adaptive immune response in the cancer immunity cycle by interacting with the T-cells. An adaptive response is developed by the APCs via the tumorigenic antigens on their surface using a major histocompatibility complex (MHC) complex in the T-cells. However, mostly due to various enzymatic actions, these antigens lose their immunogenicity, and cannot enter the APCs. AuNPs, due to their specificity and stability, can overcome this issue by transferring antigens directly to the APCs without enzymatic degradation and enhancing the T-cell and DC immune response.

Batus et al. showed that 10 nm-sized AuNPs conjugated with peptides stimulated the murine bone marrow macrophages by induction of TNF, IL-1β, IL-6, and NO [127]. Studies showed that individual AuNP or peptide-induced macrophage proliferation, but the conjugate suppressed the proliferation rate of the macrophages. Studies showed that the peptide pattern on the NPs has a great role in the proliferation rate rather than the length and polarity of the peptide. In another study, Fallarini et al. functionalized AuNPs with an average size of 2–5 nm with disaccharides, and these particles induced macrophage activation, T-cell proliferation, and increased IL-2 production [128]. The detailed study showed that 5 nm-sized AuNPs with a disaccharide are more prone to T-cell stimulation than monosaccharides due to the degradation of AuNPs in this stage.

AuNPs can cross the blood vessels and barrier by suitable surface modifications and can induce target-specific delivery to stimulate the T and DC cells. In one of the recent studies, Shinchi et al. showed that α-mannose decorated AuNPs can be used as a carrier for TLR7 ligands to target immune cells in both in vitro and in vivo studies. The synthetic toll-like receptor (TLR7) ligand (2-methoxy ethoxy-8-oxo-9-(4-carboxy benzyl)adenine (1V209)) and α-mannose were attached to the surface of Au using thiotic acid [129]. In vitro studies on the mouse, bone marrow showed that the production of cytokine is enhanced in the dendritic and human peripheral blood mononuclear cells than in the untreated bone marrow. In vivo studies showed that the use of IV209 nano conjugate can improve the amount of IgG2c antibody specific to ovalbumin, compared to the unconjugated material. This approach opens new strategies for the incorporation of antigens and adjuvants into AuNPs for immunotherapy. Polysaccharide-coated AuNPs possess excellent cancer immunotherapy effects by stimulating the DC and T-cells.

Cytosine-phosphate-guanine (CpG) is a potent innate stimulant for the immune system. CpG binds to the toll-like receptors which are present in the APCs to stimulate the secretion of cytokines, and CD8+ T-cell programming. Due to the excellent macrophage-clearing properties of AuNPs, the incorporation of CpG adjuvants along with cancer drugs can improve the immune responses of T and DC cells. Lin et al. showed that the incorporation of oligonucleotide-modified CpG to AuNPs can enhance the CpG delivery without any side effects and nanoparticles with a size less than 15 nm showed excellent immune response by activating the stimulatory CpG macrophages. In vivo studies in mice showed that these CpG-AuNPs can inhibit tumor growth and improve the survival rate of the mice. AuNPs impart stability for the CpG adjuvant in the tumor cycle without sacrificing the DNA tag [130]. Dendrimer-encapsulated AuNPs can be used for the efficient transfer of CpG vectors into the dendritic cells. Chen and coworkers used polyamidoamine (PAMAM)-AuNPs nanoconjugate for the delivery of CpG into the bone marrow-derived dendritic cells [131]. Since CpG possesses a negative charge on the surface, it is difficult to deliver it into the DCs. Positively charged PAMAM dendrimers can encapsulate negatively charged CpG and AuNPs, improving the biocompatibility and the transfection efficiency of the conjugate. In vitro and in vivo studies using xenografted melanoma tumor models showed that the nanoconjugate can enhance and activate the T-cells in bone marrow-derived dendritic cells to obtain improved CD4 and CD8 antigen expressions via intravenous injection than intratumoral. A similar strategy has been used for the transport of CpG-oligodeoxy nucleotides for the antigen expression ability in bone marrow-derived dendritic cells and activating the T-cells [132]. To improve the antigen loading efficiency in dendrimer-coated AuNPs, zwitterionic 2-methacryloyloxyethyl phosphorylcholine was used, and studies showed that a 44.41–48.53% increase in CpG-ODN loading and T-cell maturation occurred. Luo et al. showed that thiolate CpG-oligonucleotide bearing hollow AuNPs showed an enhanced secretion of TNFα from RAW264.7 cells by ∼15 fold than the CpG or CpG-oligonucleotide alone. Highlighting the importance of hollow AuNPs for cancer immunotherapy [133].

In another approach, AuNPs can be functionalized with cytokine-type tumor necrosis factor (TNF) for cancer immunotherapy. Paciotti et al. showed that thiolated PEG-modified AuNPs with an average size of 33 nm can be used for the delivery of recombinant human TNF in MC-38 colon carcinoma tumors. Using AuNPs resulted in a maximum antitumor response than the TNF alone, and these AuNPs-TNF nanoconjugates showed decreased off-toxicity than TNF alone [134,135]. Combined Antigen/adjuvant delivery can also be achieved using AuNPs for DC maturation and adoptive immunotherapy. Zhou et al. prepared a nano-Au cocktail in which two different AuNP systems one with CpG adjuvant, and another with OVA peptide antigen were admixed. Studies showed that the intravenously transfused nano-Au cocktail induced in vivo homing in lymphoid tissues and activated the CD8+ T-cell responses [136].

Thus, from the existing studies, it is clear that AuNPs can be used as a carrier for antigens and adjuvants for cancer immunotherapy. Suitable surface charge and surface modifications enabled AuNPs to be a better candidate for the transport of various immune-responsive antigens. The chemical inertness, stability, and biocompatibility of AuNPs under physiological and homeostatic conditions can be explored well for using them as an excellent nanocarrier in cancer immunotherapy.

#### 4.2.2. AuNPs as Nanocarrier for Cancer Vaccines

The application of cancer vaccines is another approach to cancer immunotherapy [137,138]. However, due to the low ability of these vaccines to stimulate the immune response and T-cells, their applications are limited. Moreover, the site-specific targeting and vaccine loads in tumor sites are also challenging issues for using them in cancer immunotherapy. Many of the studies showed that AuNPs can overcome most of the drawbacks of using cancer vaccines for tumor therapy. AuNPs can improve the payload, and enhance the site bioavailability of the vaccines due to their excellent size tunability properties.

In an HIV model, Arnaiz et al. showed that AuNPs with an average size of 1.8 nm, functionalized with HIV gp120 antigen can mimic oligomannosides for the maturation of DC, and via endocytosis, it inhibits the HIV infection by activating the T-cells [139]. AuNPs enabled site-specific targeting for DC via the interaction of the C-type lectin receptors on the surface of DC and the n-terminal mannose glycan clusters on HIV gp 120 using multiple Ca^2+^ interactions. This study highlighted the importance of mannose-coated AuNPs for biomimicry of the gp120 antigen and as cancer vaccines for specific maturation of DC and the activation of T-cell immunotherapeutic applications. In another study, Lin et al. showed that tumor-targeting antigen peptides can be delivered to tumor sites using AuNPs [140]. Three different major histocompatibility complex class I peptides, one from antigen model ova albumin, two from melanoma antigens, one from gp100, and one from Trp-2 were incorporated in AuNPs using carbodiimide chemistry, into a thio-PEG-AuNPs with an average size of 30 nm. This method offered 90% peptide loading efficiency and the Au nano-vaccine showed minimal cytotoxicity and stimulated the cytotoxic T lymphocytes four-fold more than the free antigen peptide vaccines. The antitumor immunogenicity for these Au nano vaccines was studied by measuring the antigen-specific CD8+ cytotoxic lymphocyte T-cells IFN-γ secretion. The maximum efficiency was obtained for the Au nano vaccines with ovalbumin and gp100 antigen peptides. In another study, polysaccharide-coated AuNPs were used for targeting APCs expressing Dectin-1 [141]. Studies showed that the polysaccharide β-1,3-glucans coated AuNPs showed in vivo immune response and stimulated the antigen-recognizing T immune cells, with an antitumor balance of expressed cytokines with a limited expression of immunosuppressive Il-10.

Cancer immune vaccines and their formulations are mainly used to induce the formation of major histocompatibility complex type I complexes, which are restricted to the CD8+ cytotoxic T-cell responses. Cao et al. prepared an anticancer vaccine from 50 nm-sized AuNPs, with surface modification using thiolated hyaluronic acid and ovalbumin antigen [142]. The photothermal property of AuNPs using near-infrared light irradiation was used for the controlled cytosolic antigen delivery and the photothermal ability of AuNPs induced antigen-specific CD8+ T-cell responses. Studies in mice showed that under laser irradiation the cytosolic antigen delivery occurred due to the endo/lysosome disruption and due to the excessive reactive oxygen species formation, enhanced proteasome activity, and downstream MHC I antigen presentation occurred, and eventually, the CD8+ T-cells are stimulated for the anticancer response.

Combined immunotherapy using cancer adjuvants and cancer vaccines using AuNPs exhibited a myriad of examples in recent years. The codelivery method induces a specific antibody response to the tumor cells by activating the antigen-specific cytotoxic lymphocyte antigen response. MUCI is an antigen-based glycoprotein that is overexpressed in many tumor tissues. Liu et al. prepared an antitumor vaccine with MUCI glycopeptide and α-GalCer, and AuNPs were used as a carrier for the delivery of this vaccine to trigger the immune response. Studies showed that this nano-vaccine approach induced enhanced tumor suppression in mice models [143]. Lee et al. synthesized an adjuvant vaccine nano complex using 7 nm-sized AuNPs for the T-cell response in lung metastatic studies using B16F10 melanoma tumor-bearing mice. The red fluorescent protein has been used as an antigen and CpG 1668 as an adjuvant for the studies. After the injection, the nano-vaccine is accumulated in the lymph nodes, interacts with the DCs, and induces a T-cell immune response via the Th1 pathway [144].

The presented literature reports suggested that AuNPs can act as a carrier for tumor vaccines with enhanced payload and specificity. AuNPs induce significant changes in the T-cell response and play a vital role in cancer immunotherapy. AuNPs-based cancer vaccines can be novel and can be used for future immunotherapeutic applications.

#### 4.2.3. AuNPs as a Nanocarrier for Antibodies

Many antibody drugs like nivolumab, pembrolizumab, cemiplimab, atezolizumab, durvalumab, and avelumab are available for cancer immunotherapy studies [145]. However, the expense, less clinical response, side effects, and high loading demand always render their usage effective for cancer immunotherapy. In most antibody-related cancer immunotherapies, activation of PD-1 and PDL-1 immune checkpoints are focused so that the native immune response in cancer cells is retained. Currently, with the aid of AuNPs, antibody transport to the tumor sites can be achieved easily by suitable surface modifications of AuNPs. AuNPs can induce target-specific delivery of antibodies due to the surface functionalization property and due to the large surface-to-volume ratio, the antibody loading efficiency can also be enhanced. PD-1 is the most targeted immune checkpoint, which is expressed in the T-cells, and it can interact with the PDL-1 ligands which are overexpressed in the tumor cells. This interaction inhibits the immune-responsive nature of the T-cells and therefore induces apoptosis to the T-cells and inhibits the immune response [146,147].

Many features like drug resistance and difficulty in the transport of PDL-1 antibodies to the tumor sites can be rectified by AuNPs, since AuNPs, due to their surface properties, pass through the tumor micro-environment barrier easily along with the conjugated antibodies and drugs. Emami et al. synthesized AuNPs and conjugated them with doxorubicin (DOX) and anti-PDL-1 targeting antibodies using lipoic acid polyethylene glycol unit with an average size of 40 nm. NIR irradiation studies showed 66% apoptosis for the CT-26 cell line. The synthesized drug delivery system enabled combined chemotherapy and immunotherapy for colorectal cancer cells [148].

AuNPs, due to their specific surface plasmon resonance properties along with the antibody immunotherapy, can be used as a contrasting agent for computed tomography imaging. Therefore, the incorporation of immune checkpoints with AuNPs can bring new theragnostic materials for cancer immunotherapy. Meir et al. showed that AuNPs conjugated with anti-PDL-1, showed excellent antitumor response with a minimum amount of the immune checkpoint ligands. Under computed tomography scanning the obtained signal from the mice bearing AuNPs gave a direct correlation between the tumor growth and the T-cell infiltration. Thus, the designed AuNPs system can be used as a measure to study the outcome of the tumor treatment and for future theragnostic applications [149].

In general, AuNPs can enhance the delivery of various antibodies to suppress the expression of PDL-1. However, detailed information is still required for the clinical trials, and many of the studies are limited to in vitro.

#### 4.2.4. AuNPs as a Nanocarrier for Genetic Drugs

Genetic drugs like siRNA and their nanoformulations enable the synthesis of various antigens or stimulate the production of various immune antigens. Such approaches find application in cancer immunotherapy. AuNPs are known for their DNA/RNA loading efficiency via suitable functional groups, a similar strategy can be used for the incorporation of siRNA into AuNPs [150]. The advantage of using AuNPs is that they can impart stability to siRNA under physiological conditions, which is otherwise a difficult task. In addition, AuNPs possess less toxicity compared to other NPs, and selective gene slicing and transfection can be achieved for siRNA [151].

Hou et al. synthesized PAMAM dendrimer entrapped AuNPs partially modified with polyethylene glycol monomethyl ether for the conjugation of pDNA/siRNA for cancer immunotherapy. The nanocomplex is used to deliver B-cell lymphoma 2 effectively to the human cervical cancer cell lines. The nanocarrier suppressed the expression of green fluorescent protein and luciferase reporter genes [152]. In another study, Labala et al. showed that a layer-by-layer assembly of AuNPs with anti-STAT3 siRNA and imatinib mesylate can be used for treating melanoma [153]. The studies on B16F10 melanoma cells showed that either the AuNPs-STAT3 siRNA or AuNPs-imatinib mesylate can induce apoptosis and decrease cell viability. The codelivery of imatinib mesylate and STAT3 siRNA using the layer-by-layer assembly of AuNPs showed a significant reduction in tumor volume, and weight, and suppressed the STAT3 protein expression drastically. Thus, this codelivery AuNPs nanocarrier system can be used for future immunotherapy studies and topical iontophoretic delivery.

Xue et al. synthesized PAMAM dendrimer (generation 5) coated AuNPs for delivering PD-L1 small interfering RNA (siPD-L1) for cancer immunotherapy [154]. The siPD-L1 suppressed the gene expression for PD-L-1, and AuNPs impart stability, cytocompatibility, and effective delivery of this gene to the tumor sites. The immunotherapeutic application of this nanocarrier was studied by enhanced expression of CD8+ and CD4+ T-cell infiltration in spleen and cancerous tissue. The T-cell activation programmed the immune response of the tumor cells and induced cancer immunotherapy using the siRNA-AuNPs delivery system. In another study, Gulla et al. showed that peptide coated AuNPs in conjugation with immune checkpoint inhibitor (PD-L1siRNA ) and signal transducer and activator of transcription 3 inhibitor showed high survival rate of 70%, in melanoma bearing mice than the untreated mice [155].

Liu et al. loaded the siRNA-PDL-1 complex into Au nanoprisms for targeted delivery of the SiRNA-PDL-1 complex to lung cancer sites (Figure 8a) [156]. For the effective adsorption of negatively charged siRNA, the surface of Au nano prisms was coated with poly (sodium 4-styrenesulfonate) (PSS) and poly (diallyldimethylammonium chloride (PDADMA). Both in vivo and in vitro studies showed that the synthesized nanocarrier can effectively deliver siRNA-PDL-1 complex and showed suppression in the expression of hPD-PDL-1 in HCC827 cells (Figure 8b,c). Due to the inherent optical properties of Au nanoprisms, the formed nano complex is used for photoacoustic imaging and photothermal therapy for lung cancer (Figure 8d).

### 4.3. AuNPs for Photothermal Cancer Immunotherapy

In recent years, cancer immunotherapy has become an important strategy for cancer treatment [157,158,159]. Preclinical studies have shown that cancer immunotherapy can overcome the failures in conventional cancer treatment methods for primary and metastatic tumors. In treatment using drugs, the innate immune systems of the human body are activated toward the tumor cells and enhance the effector cell number, along with reducing the host’s suppressor mechanisms. Nevertheless, many immunotherapy systems lack target specificity, and some of them induce unwanted side effects [160,161]. To avoid this problem, currently, AuNPs are used as a stimulus and target specific carriers for cancer cells to stimulate the immune response system of the host body [162]. Many immunotherapeutic agents can be incorporated into the AuNPs, and studies have shown that nano-immunotherapy can enhance the anticancer activity of the drugs and increase the survival rate of many cancer patients [163,164].

For cancer immunotherapy, the PT effect of AuNPs can be used to secret heat shock proteins, antigens, pro-inflammatory cytokines, etc., which then activate the T-cell for an anticancer immune response [165,166]. AuNPs can induce direct cancer cell death and activate immunogenic cell death (ICD). Although a synergic effect of PTT and ICD can improve the tumor treatment strategies using AuNPs, the optimum thermal dosage to be applied is still under consideration (Figure 9) [167]. A high or low thermal dosage will not induce an ICD, and the use of AuNPs alone cannot induce an ICD effectively. AuNPs can be used directly as an immune-stimulating agent; alternatively, a combination strategy using various immunoadjuvants, immune check point inhibitors, and combined chemotherapy along with PTT is a strategy recently investigated extensively.

#### 4.3.1. ICD Directly Induced by AuNPs

The PT effect induced by AuNPs generates damage-associated molecular patterns, therefore generating various immune-responsive pathways for tumor treatment.

Ma et al. showed that the self-assembly of AuNPs in liposomes induced a redshift in their absorption properties to the NIR-II region (Figure 10a) [168]. Optimization of the size of AuNPs and the composition of liposome induced a maximum redshift and NIR-II absorption band at 964 nm for Au40C-DOPC (SPR peak at 964 nm) with a PT conversion efficiency of 21.88% (Figure 10b). In vitro studies showed that PTT-induced ICD induced the generation of damage-associated molecular patterns (DAMPs). By contrast, a more homogenous generation and distribution of DAMPs was observed in vivo, resulting in both innate and adaptive immune responses under the NIR-II region with efficient PTT for tumor therapy by DAMPs and ICD (Figure 10c,d). The NIR-II laser irradiation allowed the production of IFNγ-producing CD4+/CD8+ T-cells and natural killer (NK) cells via the DC maturation. The AuNP self-assembly method enables an effective strategy for cancer immunotherapy using PTT for distal and metastatic cancer.

#### 4.3.2. PTT Combined with Immunoadjuvants

The PT therapy of AuNPs with immunoadjuvants is an approach for cancer immunotherapy. Attaching various immunoadjuvants to AuNPs induces the maturation of DCs and T-cell activation in many of the tumor-bearing cells. Zhou et al. showed that bioinspired BSA-coated AuNRs with an average diameter of 122.1–11.6 nm, containing cetyltrimethylammonium bromide and immunoadjuvant imiquimod, can induce a significant immune response by PTT for melanoma ablations [169]. The PT therapy-induced secretion of various immune-responsive cytokines such as TNF-a, IL-6, and IL-12. The immunoadjuvant along with the PTT induced by AuNR inhibits the tumor growth and activation of the T-cells and DCs by 65.1% with activation of the memory immune systems. Zhang et al. prepared an AuNS delivery system with siRNA and adjuvant CpG, and their study showed that the nanocarrier system possessed excellent tumor ablation and immune response toward gastric tumor cells [170]. The nanomaterials were evenly distributed in the cancer and killer tumor cells in the body. The AuNS material caused almost 66.9% of apoptosis under 5 min of laser irradiation with a long survival rate for the tumor-bearing mice. Furthermore, flow cytometric studies showed that the nanomaterial system induced DC maturation via CD80+CD86+ generation up to 66% compared to the control PBS buffer, along with increased levels of IL-2 and IL-6.

In another study, Chen et al. showed that polyethyleneimine-protected AuNR could be complexed with CpG adjuvant, and it showed excellent PT and immunotherapy response for 4T1 cells [171]. Laser irradiation studies revealed that the nanomaterial showed an increase in early and late apoptotic cells to 28.4% and 70.4%, respectively. Furthermore, the irradiation studies showed a significant DC infiltration via CD11c+ up to 14.78% compared to the control PBS, which was 8-fold higher than the saline-treated cells after PTT. Yata et al. prepared an immunostimulatory hydrogel from AuNPs with adjuvant CpG and DNA, which induced secretion of antitumor inflammatory mediators such as TNF-α, IL-6, and IFN-γ in HSP70 in tumor tissue in mice by PTT [172]. Studies showed that this potential nanoparticle system could be used for combined PTT and immunotherapy studies with enhanced survival rates for mice.

#### 4.3.3. PTT Combined with Immune Checkpoint Inhibitors

Cancer immunotherapy with specific immune checkpoint inhibitors offers an excellent platform for cancer immunotherapies. The two important immune check points are cytotoxic T lymphocyte antigen-4 (CTLA-4) and programmed cell death protein 1 (PD-1)/programmed cell death-ligand 1 (PD-L1). Therefore, blocking these two immune checkpoints can enhance cancer immunotherapy. Studies showed that blocking the CTLA-4 immune check points could activate the T-cells and reduce the immune-suppressive nature of T-cells in the tumor micro-environment [166].

Various studies have shown that the combination of AuNPs and the PD-1/PDL-1 immune checkpoints is a well-known strategy for PTT-induced immune therapy. Liu et al. showed that administration of the PD-L1 immune checkpoint blockade in AuNSTs induced excellent immune PTT modality for MB49 bladder cancer cells in mice Figure 11a [173]. As shown in Figure 11b, the PT property of AuNSTs induces heat in the primary tumor cells in MB49 bladder cancer cells by laser irradiation and kills the primary tumor cells. Incorporation of the PDL1 immunoblockade in one arm of AuNSTs induced immunological response along with the PT effect of AuNSTs and allowed growth suppression of the distant tumor without laser application, demonstrating the concept for cancer immunotherapy (Figure 11c). The studies indicated that after 7 days of post-injection, the combined treatment showed an increased level of T, CD4, CD8 T, and B cells in the spleen, whereas the number and amount of myeloid-derived suppressor cells were reduced drastically. The PD-1 expression on CD4 and CD8 T-cells was up-regulated using the combined immune therapy studies with an increase in the PD-1^+^CD4 and PD-1^+^CD8 T-cells and an increased survival rate for the mice (Figure 11d). In another study, Odion et al. showed that AuNSTs could induce the release of tumor-associated antigens, heat shock proteins, and DAMPs. The PT irradiation and a combination of anti-PDL-1 checkpoint inhibitors could induce a high survival rate in a group of mice bearing tumor cells [40].

Yang et al. showed that the Au@Pt NP system with a rationally designed peptide conjugate afforded excellent cancer PT immunotherapy applications [174]. The peptide design induced the generation of a D-peptide antagonist of programmed cell death-ligand 1 (PD-L1) during the PTT. The in vivo studies in 4T1 breast cancer showed an enhancement in T-cell generation by the blockage of PD-L-1 immune checkpoints to cure primary tumors and decimate lung metastasis. In a report, Cheng et al. showed that AuNCs could be used as a multifunctional nanomaterial for cancer immunotherapy [175]. The AuNCs loaded with ansamitocin P3 (AP3) and anti-PDL1 binding (AP3-AuNCs-anti-PDL1) showed excellent immune response by irradiation with an NIR light source for 10 min. Highly activated DCs were generated by the controlled release of AP3 with enhanced T-cell proliferation. The present AP3-AuNCs-anti-PDL1 nanocarrier system strengthens cancer immune therapy studies, particularly for hepatocellular carcinoma. In another study, Luo et al. showed hollow AuNS coated with poly(d, l-lactic-co-glycolide) loaded with anti-PD-1 peptide eliminated most of the primary tumors and suppressed the growth of distant uninfected primary tumors, with long survival time for the mice by the generation of CD8+ T-cells [176]. Enhanced levels of CD8+ and CD4+ T-cells were observed in the spleen and peripheral blood mononuclear cells in the mice after treatment.

Therefore, the PT effect of AuNPs plays a pivotal role in advancing the field of cancer immunotherapy. By harnessing the unique properties of AuNPs, such as their efficient light absorption and conversion into localized heat, researchers have unlocked a powerful strategy to synergistically enhance the immune response against cancer cells. Upon irradiation with NIR light, AuNPs generate controlled hyperthermia within the tumor micro-environment. This localized heat not only induces direct tumor cell destruction but also triggers a cascade of immunomodulatory events. The heat shock response triggered by PTT leads to the release of tumor-associated antigens and pro-inflammatory cytokines, activating dendritic cells and other immune cells crucial for initiating an antitumor immune response. This orchestrated interplay between the PT effect and immune modulation holds immense promise for overcoming the immune evasion mechanisms employed by cancer cells, ultimately leading to more effective and durable outcomes in cancer immunotherapy. The application of the PT effect of various AuNPs in cancer immunotherapy is listed in Table 2.

#### 4.3.4. PTT-Based Combinatorial Treatments

In addition to adjuvants, antibodies, and immune checkpoint blockades, AuNPs can be incorporated with many drugs and site-specific receptors for targeted PTT immunotherapy. Nam et al. showed that PTT along with chemotherapy is an excellent platform for enhanced cancer immune therapy studies [177]. Polydopamine-coated spiky AuNPs were loaded with DOX and showed excellent antitumor activity and survival rate >85% in the CT26 colon carcinoma model. PTT studies showed an enhanced AH1-specific CD8^+^ T-cell response. The ability of DOX to induce DC maturation and the PTT effect of Au enhanced the chemo-PT platform for cancer immunotherapy.

HA conjugated to AuNPs can allow target-specific delivery of AuNPs and drugs to cancer sites and induce PTT [178,179]. Studies showed that AuNPs could act as a nano-vaccine with HA, induce damage to the tumor cells under NIR radiation, and activate the HA-targeting receptor-mediated endocytosis. Interaction of HA with CD44 allowed increased uptake of AuNPs for efficient PTT. In another study, Cao et al. showed that HA- and ovalbumin-coated AuNPs exhibited an enhanced PTT effect and the stimulation of DC for the production of CD8+ T-cells [26,142]. Because of the excellent NIR absorption and thermal properties of these AuNPs, disruption of endo/lysosomes induced cytosolic antigen delivery and the enhanced ROS production, augmenting the proteasome activity along with a downstream in the major histocompatibility complex class I antigen expression with potent antitumor immune response.

Liu et al. prepared AuNST@CaCO_3_/Ce-6 nanoparticles and loaded them into human NK cells [180]. Since CaCO_3_ is highly biocompatible mixing them with AuNSTs improved the biocompatibility, and AuNSTs possess excellent PTT efficiency. The nanocomplex showed enhanced PT effect and ROS formation with target specificity due to the NK cells. Moreover, Ce-6 is known for its NIR absorption and photosensitization properties, admixing it with the nano-complex-stabilized material for cellular uptake. After the NIR irradiation, the cytokine levels of tumor cells (A549) increased owing to the generation of CD3− CD56+ (92%), CD56+ NKG2D+ (98%), CD56+ 2B4+ (99%), CD56+ NKp30+ (96%), CD56+ NKp44+ (91%), and CD56+ NKp46+ (86%). This combined strategy induced excellent immunotherapy results in both in vitro and in vivo studies.

Together, various combination strategies using AuNPs-PTT and AuNPs-PTT-chemo in the field of immunotherapy are promising therapeutic strategies for treating cancer. Even though PTT and PDT are exclusively known for the phototherapy modes for cancer treatment, they possess certain drawbacks such as target specificity, lack of deep tissue penetration, and recurrence of the tumor, which is a challenging issue. Combined PTT and PDT therapies using AuNPs in the cancer immunotherapy fields are interesting. Liu et al. reported that a biocompatible gold nanocluster from captopril, Au25(Capt)18, showed excellent PT stability and enhanced ^1^O_2_ generation [181]. NIR irradiation studies on cutaneous squamous cell carcinoma showed that the contribution for cytotoxicity in the cancer cells by PTT and PDT was 28.86% and 71.14%, respectively. The nanocluster + NIR laser studies showed that, along with PTT/PDT, there was production of CD4^+^ T and CD8^+^ T-cells at 44 °C for an irradiation time of 5 min and opened a new platform for immunotherapy. Jin et al. showed that under NIR laser irradiation, a corn-like Au/Ag nanorod in conjugation with CTLA4 induced simultaneous PDT and PTT therapy along with cancer immune response in 4T1 tumor cell lines. Long-term immune memory was achieved for the mice with the generation of CD3+CD8+CD62L−CD44+ T-cells [182]. 

Recently, endoplasmic reticulum (ER) stress has been found to be a key step for the ICD using NPs. Because of the hypoxia conditions, most of the PSs target the ER, and the generation of ROS by photosensitizers creates ER stress, eventually leading to cell destruction and the ICD pathway opens up. Li et al. prepared a combined PDT/PTT-based nanomaterial with a hollow Au nanosphere and functionalized it with an ER-targeting pardaxin peptide linked to indocyanine green [183]. To reverse the hypoxia condition, oxygen-delivering hemoglobin liposome was also incorporated into the nanoparticle system. Light irradiation-induced and created ER stress along with the production of calreticulin, a biomarker for ICD. Owing to the maturation of DCs, a series of immune response cycles started in the CT-26 tumor model and generated CD8+ T-cell proliferation and cytokine secretions for cancer immune response. Moreover, in the B1 6 tumor model, the DC maturation induced increased secretion of CD11c+/CD80+/CD86+ T-cells.

## 5. Conclusions and Future Directions

In conclusion, a lot of progress has been made in the utilization of AuNPs in nanomedicine based on their unique properties. The tunable optical properties, biocompatibility, and robust surface chemistry will be the main reasons for the wide utility of AuNPs. The SPR property of Au thin film was readily adopted as a robust and essential sensing platform in the development of new drug candidates or the study of molecular interactions. The strong PT effect of AuNPs can now open a new technology in diagnostics and therapeutic applications. Now, the synthetic methods for diverse nanostructures of AuNPs are well established for high quality on a large scale. Versatile tools for further improvement of the PT effect of AuNPs were also developed and understood the mechanism. The strong PT effect of AuNPs could improve the efficiency of photoacoustic imaging, and AuNPs could be used as a thermal amplifier in PCR, which enabled the presence of portable PCR.

Beyond the conventional hyperthermia, the controlled PT effect of AuNPs showed promise in cancer immunotherapy by serving as drug carriers as well as a platform for target-specific therapy. Additionally, incorporating AuNPs into various nanoformulations such as liposomes and poly(lactic-co-glycolic acid) can further expand the therapeutic potential of AuNPs in cancer immunotherapy. Overall, AuNPs represent a promising avenue for disease detection and cancer immunotherapy, with ongoing research likely to yield valuable clinical applications in the future.

## Figures and Tables

**Figure 1 pharmaceutics-15-02349-f001:**
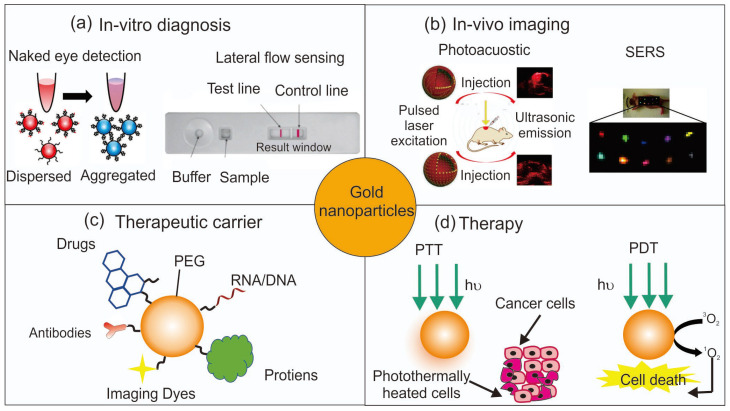
Developed diagnostic and therapeutic applications of AuNPs for (**a**) in vitro diagnosis, (**b**) in vivo imaging, (**c**) therapeutic carrier, and (**d**) phototherapy. Reproduced with permission from [14,15,16,17,18]. Copyright 2021 American Chemical Society; Copyright 2020 Wiley-VCH; Copyright 2009 National Academy of Sciences, USA; Copyright 2015 Wiley-VCH; Copyright 2017 MDPI publisher.

**Figure 2 pharmaceutics-15-02349-f002:**
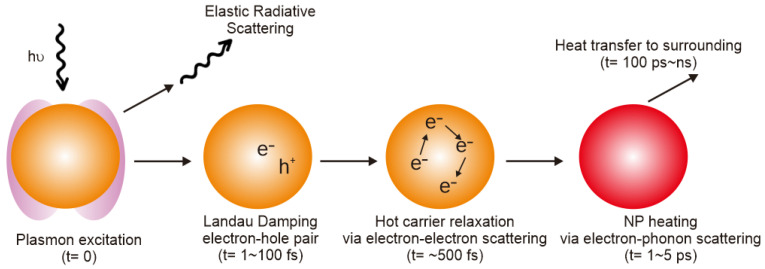
The mechanism of light-driven photothermal effect. Reproduced with permission from [42], Copyright 2019, American Chemical Society.

**Figure 3 pharmaceutics-15-02349-f003:**
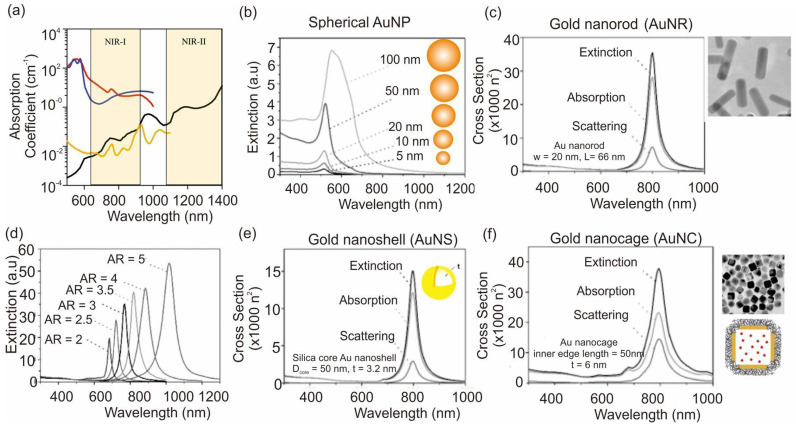
(**a**) Near-infrared window of body tissue, (**b**) calculated extinction spectra of spherical AuNPs, (**c**) AuNRs, (**d**) extinction spectra of AuNRs with increased aspect ratio, (**e**) AuNS, and (**f**) AuNC. Reproduced with permission from [44,45]. Copyright 2006, Royal Society of Chemistry; Copyright 2018 American Chemical Society.

**Figure 4 pharmaceutics-15-02349-f004:**
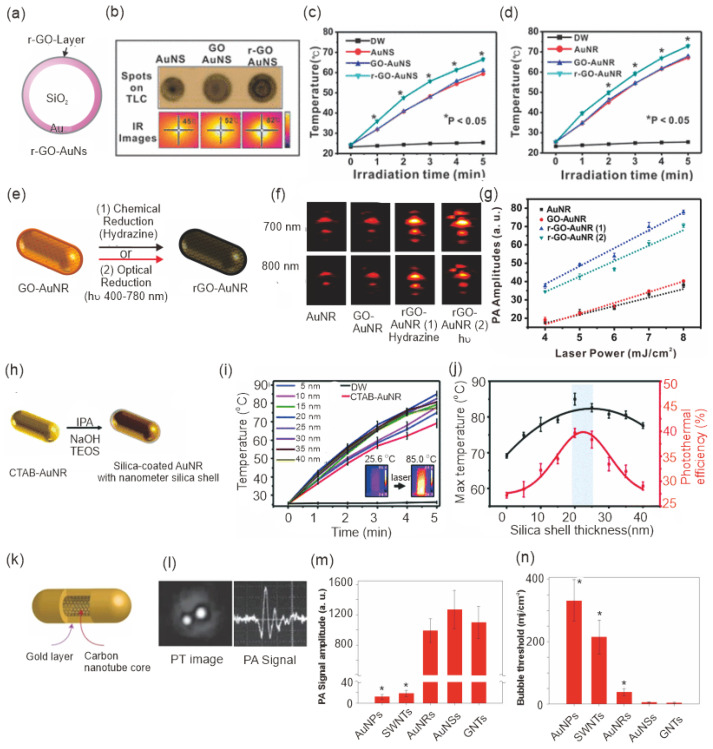
Design strategy for enhanced PT and PA imaging in AuNPs. (**a**) Design of r-GO-coated AuNS, (**b**) Thermal images of GO- and r-GO-coated AuNS using laser source treatment (3.0 W/cm^2^, 808 nm). (**c**,**d**) Temperature rise in AuNSs and AuNRs after GO and r-GO coating. (**e**) Synthesis of r-GO-AuNR via hydrazine (method 1) and light-mediated reduction (method 2). (**f**) PA images and (**g**) signal amplitude for AuNR, GO-AuNR, and r-GO-AuNR (laser power = 6.2 mJ/cm^2^; pulse rate = 10 Hz). (**h**) AuNR synthesis with controlled silica shell thickness. (**i**) Temperature variation and (**j**) photothermal efficiency for AuNR with varying SiO_2_ thickness. (**k**,**l**) Gold-plated carbon nanotube (GNT), photothermal image, and PA signal using 850 nm laser. (**m**,**n**) Comparison of PA signal intensity. * *p* < 0.05, compared to GNTs for 10 measurements. Reproduced with permission from [61,62,63,65] Copyright @ 2013, 2015, American Chemical Society; Copyright @ 2022, RSC Publishing; Copyright 2009, Nature Publishing Group.

**Figure 5 pharmaceutics-15-02349-f005:**
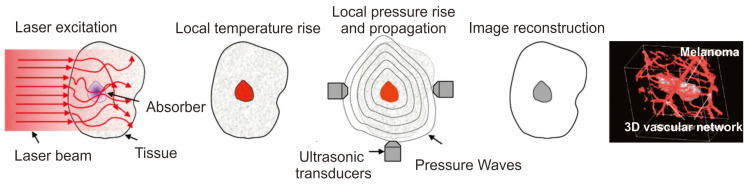
Schematic illustrating the working principle of PAT, and the high-resolution in vivo imaging of melanoma. Reproduced with permission from [70,73], Copyright 2006 and 2016. Nature Publishing Group.

**Figure 7 pharmaceutics-15-02349-f007:**
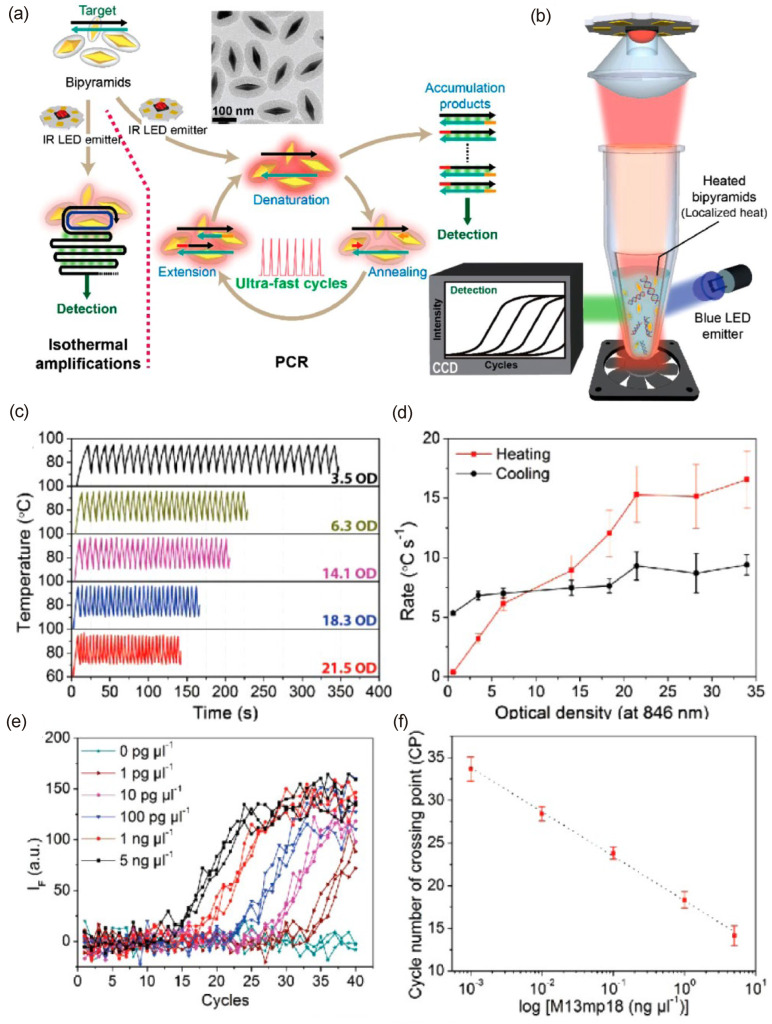
(**a**) Schematic representation of the AuNBP-based PCR amplification strategy. (**b**) Associated blue LED system for the amplification of the PCR-based system. (**c**) Temperature profile obtained for amplification of 30 cycles with different concentrations or OD of AuNBPs and (**d**) heating and cooling rates calculated for the AuNBPs using a blue LED. (**e**) Concentration-dependent amplification study of the M13mp18 DNA template. (**f**) Linear plot of the cycle number crossing points for the DNA amplification with the concentration of M13mp18 DNA. Reproduced with permission from [90]. Copyright 2017, American Chemical Society.

**Figure 8 pharmaceutics-15-02349-f008:**
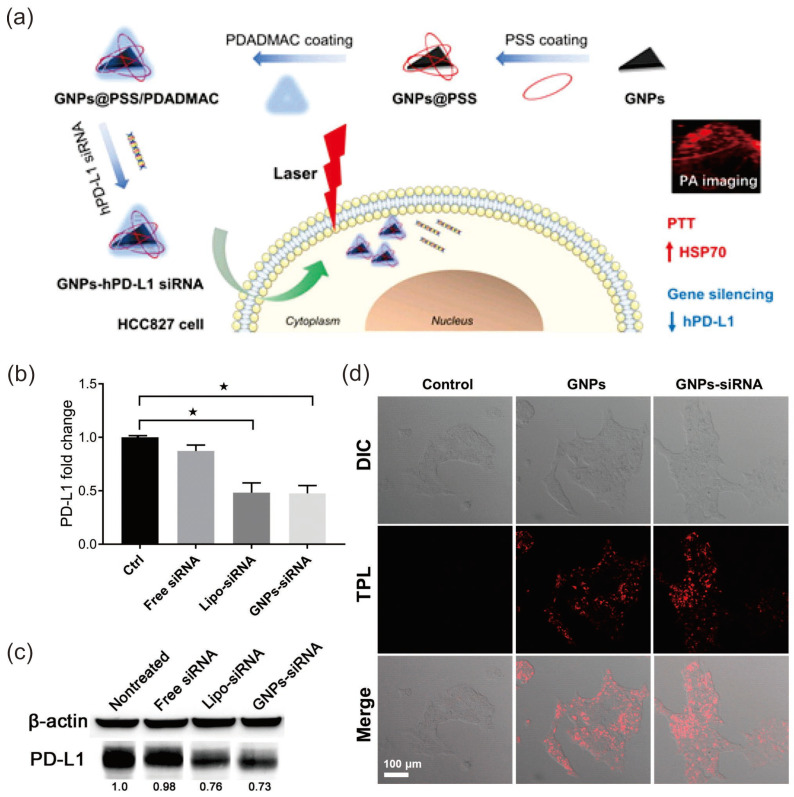
(**a**) A schematic representation of the Au nanoprism-based nanocarrier for siRNA-PDL-1 complex and their application in cancer immunotherapy. (**b**) mRNA and (**c**) protein expressions of hPD-L1 in HCC827 cells after the laser irradiation using PBS as a control (^★^
*p* < 0.05). (**d**) Confocal microscopic images for HCC827 cells treated with AuNPs and AuNPs-siRNA using 630 nm laser. Reproduced with permission from [156]. Copyright, 2019, Elsevier.

**Figure 9 pharmaceutics-15-02349-f009:**
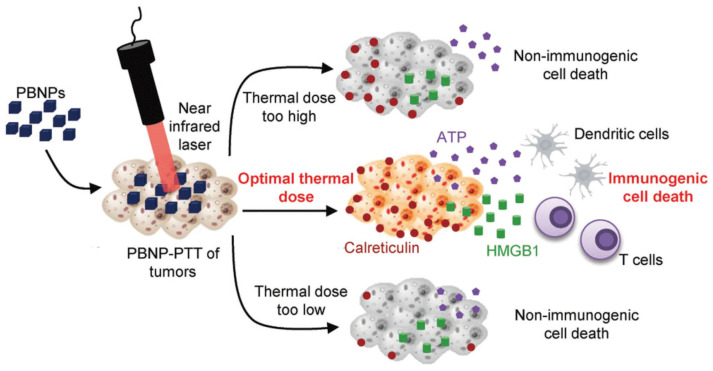
Schematic representation of the Prussian Blue-based nanoparticles showing the optimum thermal dosage for cancer immune therapy. Reproduced with permission from [167]. Copyright 2018, Wiley-VCH.

**Figure 10 pharmaceutics-15-02349-f010:**
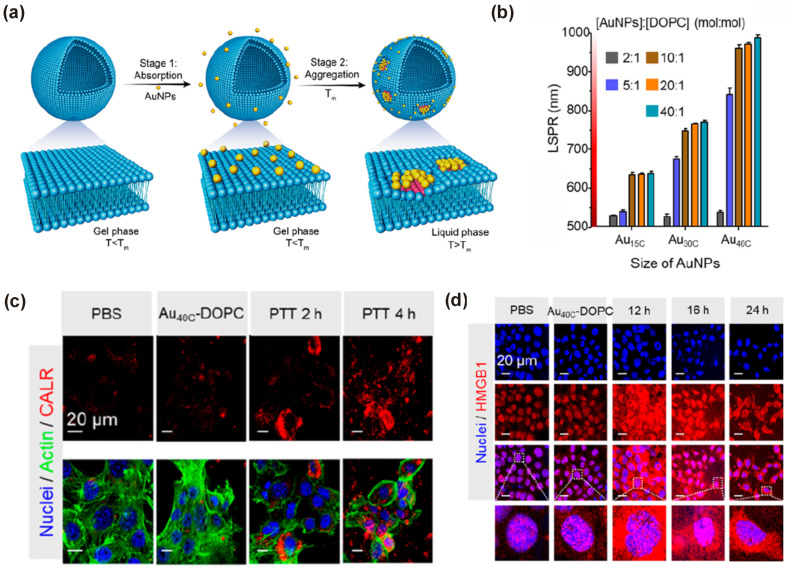
(**a**) Cartoon representation of the liposome-based self-assembly of AuNPs, (**b**) changes in the absorption peaks of AuNPs with varying composition of AuNPs and liposomes, (**c**) confocal microscopic images for 4T1 tumor cells using calreticulin after PTT treatment, and (**d**) immunofluorescence staining studies for HMGB1 expression of cancer cells after PTT. Reproduced with permission from [168]. Copyright 2019, American Chemical Society.

**Figure 11 pharmaceutics-15-02349-f011:**
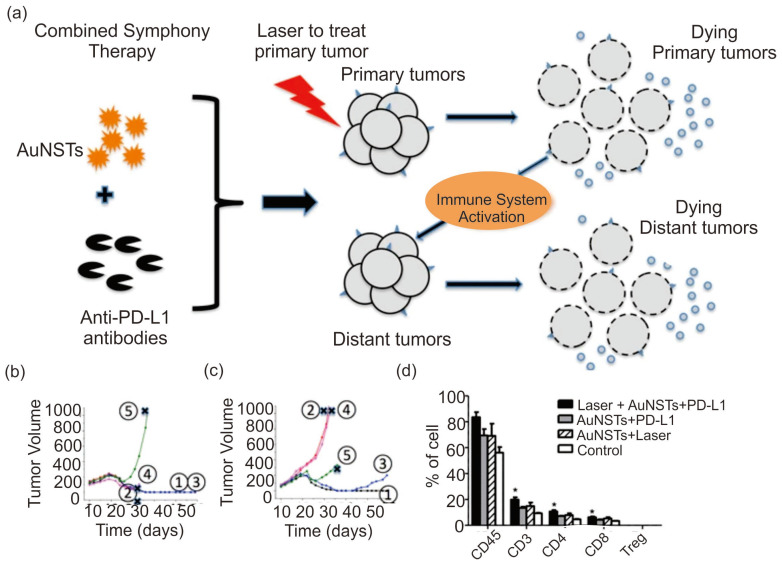
(**a**) Cartoon representation of the combined approach for immunotherapy. (**b**,**c**) Tumor growth in primary and distant tumors, respectively, using the combined approach (AuNSTs + Laser + Anti-PD-L1) in various mice (1–5). (**d**) Percentage of total leukocytes (CD45), total T-cells (CD3), CD4, CD8, and T regulatory cells (CD4/CD25/FOXP3) after the treatment. * *p*  <  0.05, compared to control group. Reproduced with permission from [173]. Copyright 2017, Springer Nature.

**Table 1 pharmaceutics-15-02349-t001:** The PT performance comparison of Au nanostructure depends on the size and shape.

Type of Au Nanostructure	Absorption Wavelength (nm)	Size of Gold	PT Efficiency	Excitation Wavelength	Ref.
Rod shape	805	17 × 56 nm	22%	0.4 W/cm^2^, 808 nm	[48]
Rod shape	770	7.4 × 25.7 nm	50%	CW laser, 2 W/cm^2,^ 808 nm	[49]
Cage shape	800	45 nm edge length, 5 nm wall thickness	64%	0.4 W/cm^2^, 808 nm	[50,51]
Spherical shape	532	20 nm	97–103%	0.28 CW laser, 532 nm	[52]
Nanoshell	815	50 nm	59%	815 nm	[53]
Nanomatryoshkas	560, 783	152 nm	39%	CW laser, 2 W/cm^2^, 809 nm	[54]
Nanoflower	550–800	145 × 123 × 10 nm	74%	1 W/cm^2^, 809 nm	[55,56]
Nano ring	1058	400 nm	56%	CW laser, 0.1 W/mm^2^, 700–900 nm	[57]
Core shell	809	10–77 nm	51–95%	CW laser, 809 nm	[58]

**Table 2 pharmaceutics-15-02349-t002:** List of AuNP structures combined with immunoadjuvants or checkpoint blockades for photothermal immunotherapy.

Photothermal Nanoparticles	Immunoadjuvants or Checkpoint Blockade	Effector Cells	Cytokines	Tumors	References
(BSA-AuNRs)	R837	DCs, CD8+ T-cells	TNF-α, IL-6, IL-12	Murine melanoma cell B16-F10	[139]
AuNSs	CpG	DCs, CD8+ T-cells, CD4+ T-cells	IL-2, IL-6, IFN-γ	Murine gastric cancer cell MFC	[140]
AuNR-PEI	CpG	DCs, CD8+ T-cells, CD4+ T-cells	-	Murine breast cancer cell 4T1	[141]
AuNR-DNA hydrogels	CpG	-	TNF-α, IL-6, IL-12p40, IFN-γ	Murine T lymphoma cell EG7-OVA	[142]
AuNSTs	Anti-PD-L1	CD8+ T-cells, CD4+ T-cells, B cells	-	Murine bladder cancer cells MB49	[143]
AuNSTs	Anti-PD-L1	CD45), (CD3), CD4, CD8, and T regulatory cells (CD4/CD25/FOXP3)	-	Brain tumor	[144]
Au@Pt NPs	Anti-PD-L1	CD8+ T-cells, CD4+ T-cells	TNF-α, IL-6, IL-12p70, IFN-γ	Murine breast cancer cell 4T1	[145]
AuNCs	Anti-PDL1	CD11c, CD80, CD11c CD86	-	Hepatocellular carcinoma	[146]
HAuNS	Anti-PDL1	DCs, CD8+ T-cells	TNF-α, IL-2, IL-12p70, IFN-γ	Murine breast cancer cell 4T1, murine colon cancer cell CT26	[147]

## Data Availability

Not applicable.

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
