# Peer review of "Photothermal Effect of Gold Nanoparticles as a Nanomedicine for Diagnosis and Therapeutics"

_pharmaceutics, 2023, doi:10.3390/pharmaceutics15092349_

Round 1

Reviewer 1 Report

The review article by Kumar et al. provides a comprehensive overview of the utilization of gold nanoparticles (AuNPs) in photothermal-based diagnosis and therapeutic applications. The authors discuss a range of strategies aimed at enhancing the photothermal efficiency of AuNPs. These efforts are subsequently explored in their applications in photoacoustic imaging and plasmonic PCR-based nucleic acid detection. Notably, the article uniquely emphasizes the application of AuNPs in photothermal-immunotherapy for cancer treatment, setting it apart from existing literature reviews.

However, the abstract and introduction require refinement to clearly establish the distinctiveness and significance of this review within the scientific community. Additionally, the authors are encouraged to incorporate a dedicated section elucidating how varying sizes and shapes of AuNPs contribute to different levels of photothermal efficiency.

While previous review articles have covered AuNPs' role in cancer photothermal therapy (PTT), it is suggested that the authors focus more extensively on the PTT-based immunotherapy of cancer, a relatively underexplored aspect that could provide greater insight.

To enhance readability and comprehension, the inclusion of a graphical representation or table outlining different approaches employed in AuNPs-based photothermal-immunotherapy would greatly benefit readers.

Lastly, a concise summary detailing the mechanism underlying AuNPs-based photothermal-immunotherapy would serve as a valuable takeaway for readers, offering a clearer understanding of the discussed concepts. These revisions would strengthen the article's contribution and render it suitable for consideration in the Pharmaceutics journal.

Minor editing of English language required.

Author Response

Reviewer 1: The review article by Kumar et al. provides a comprehensive overview of the utilization of gold nanoparticles (AuNPs) in photothermal-based diagnosis and therapeutic applications. The authors discuss a range of strategies aimed at enhancing the photothermal efficiency of AuNPs. These efforts are subsequently explored in their applications in photoacoustic imaging and plasmonic PCR-based nucleic acid detection. Notably, the article uniquely emphasizes the application of AuNPs in photothermal-immunotherapy for cancer treatment, setting it apart from existing literature reviews.

Q1: However, the abstract and introduction require refinement to clearly establish the distinctiveness and significance of this review within the scientific community. Additionally, the authors are encouraged to incorporate a dedicated section elucidating how varying sizes and shapes of AuNPs contribute to different levels of photothermal efficiency.

Answer): The abstract and the introduction has been modified in the revised manuscript.

The query related with the size and shape dependant photothermal property of AuNPs, we have included a detailed description in session 2 with the basic principle of photothermal effect of AuNPs and included a description in the form of table 1 in the submitted manuscript. 

Q2: While previous review articles have covered AuNPs' role in cancer photothermal therapy (PTT), it is suggested that the authors focus more extensively on the PTT-based immunotherapy of cancer, a relatively underexplored aspect that could provide greater insight.

Answer): Thank you for the valuable suggestion. We have included this part in the revised manuscript.

Q3:To enhance readability and comprehension, the inclusion of a graphical representation or table outlining different approaches employed in AuNPs-based photothermal-immunotherapy would greatly benefit readers.

Answer): We have included a table -- in the revised manuscript outlining the different approaches employed in AuNPs-based photothermal-immunotherapy.

Q4: Lastly, a concise summary detailing the mechanism underlying AuNPs-based photothermal-immunotherapy would serve as a valuable takeaway for readers, offering a clearer understanding of the discussed concepts. These revisions would strengthen the article's contribution and render it suitable for consideration in the Pharmaceutics journal.

Answer): Thank you for the valuable suggestion. We have included a summary part in the revised manuscript highlighting the mechanism of AuNP-based photothermal-immunotherapy.

Reviewer 2 Report

This review highlights recent advancements in gold nanoparticle (AuNP)-based diagnostic and therapeutic applications, primarily centered on the photothermal (PT) effect. While the PT effect of AuNPs has been explored for hyperthermia, photodynamic therapy, and sensitive imaging like photoacoustic tomography, their applications are expanding into new diagnostic and therapeutic domains requiring innovative material design and approaches. AuNPs' size and shape influence their PT-conversion efficiency, enabling applications in nucleic acid detection and serving as thermal amplifiers for rapid heating and cooling. Additionally, AuNPs are potential carriers for immune vaccine delivery, activating antitumor immune responses, and offering promising prospects for PT-based cancer immunotherapy to eliminate primary and distant tumors while preventing recurrence. The review emphasizes recent progress in PT effect-based applications using novel strategies and AuNP-based material designs. In general, the manuscript was well-written and the article structure was clear and the design was rigorous. I suggest accepting with only some proofreading for language.

While the manuscript is generally well-written, I recommend thorough proofreading for language and grammar.

Author Response

Thank you for considering our review for the publication. We performed the proofreading and revised.  

Reviewer 3 Report

This review paper entitled: Photothermal effect of gold nanoparticles as a nanomedicine for diagnosis and therapeutics presents the theory and uses of photothermal effects in the diagnosis treatment of diseases. The discussion of the theory which underpin the photothermal effects of the variety of gold nanoparticles is very useful for the novice and acts as refresher for practitioners. The presentation is easy to follow, and the use of the diagrams and tables is effective. However, the material can be better presented into four clearly demarcated sections to cover the basics, detection, therapeutics, and conclusion. The current numbering of the sections and sub-sections should be re-visited to ensure that they are logical. The legend to Figure 4 is lengthy should be reduced. The conclusions and future developments are too long and should focus on the key areas of potential developments. The length of the paper is commensurate with the material being presented. There are a few errors in expression. Overall, the paper is well written and informative both for the expert and beginner.

The quality of English is good and the material presented is easy to  understand and require only minor editing in parts.

Author Response

Reviewer 3: This review paper entitled: Photothermal effect of gold nanoparticles as a nanomedicine for diagnosis and therapeutics presents the theory and uses of photothermal effects in the diagnosis treatment of diseases. The discussion of the theory which underpin the photothermal effects of the variety of gold nanoparticles is very useful for the novice and acts as refresher for practitioners. The presentation is easy to follow, and the use of the diagrams and tables is effective.

Q1: However, the material can be better presented into four clearly demarcated sections to cover the basics, detection, therapeutics, and conclusion. The current numbering of the sections and sub-sections should be re-visited to ensure that they are logical.

Answer): Thank you for the comment. We revised accordingly.

Q2: The legend to Figure 4 is lengthy should be reduced.

Answer): We have modified the figure caption and reduced the length.

Q3: The conclusions and future developments are too long and should focus on the key areas of potential developments.

Answer: The conclusion and future development part has been modified in the revised manuscript.

Q4: The length of the paper is commensurate with the material being presented. There are a few errors in expression. Overall, the paper is well written and informative both for the expert and beginner.

Answer: We have taken care the issue related with expressions.

Round 2

Reviewer 1 Report

The authors have modified the manuscript based on the reviewer's comments, resulting in significant improvement of the manuscript. I recommend acceptance of the manuscript. 

Minor editing of English language required